# Endowing Pre-trained Graph Models with Provable Fairness

## ABSTRACT

Pre-trained graph models (PGMs) have received considerable attention in graph machine learning by capturing transferable inherent structural properties and applying them to different downstream tasks. Similar to pre-trained language models, PGMs also inherit biases from human society, resulting in discriminatory behavior in downstream applications. However, the debiasing process of most existing methods is coupled with parameter optimization of GNN, making them not efficient to debias PGMs. Moreover, these debiasing methods lack a theoretical guarantee, i.e., provable lower bounds on the fairness of model predictions, which directly provides assurance in a practical scenario. To overcome these limitations, we propose a novel framework that endows pre-trained **Graph** models with **P**rovable f**AiR**ness (called GraphPAR). GraphPAR freezes the parameters of PGMs and applies a parameter-efficient adapter on node representations to make the model's predictions fairer. Specifically, we design a sensitive attribute augmenter that extends node representations with different sensitive attribute semantics for each node. Then employ two adversarial debiasing methods to optimize the adapter's parameters. Furthermore, based on the proposed framework GraphPAR, we quantify whether the fairness of each node is provable fairness, i.e., predictions are always fair within a certain range of sensitive attribute semantics. Experimental evaluations on real-world datasets demonstrate that GraphPAR achieves state-of-the-art performance and fairness on node classification task. Furthermore, based on our GraphPAR, around 90% nodes have provable fairness.

## CCS CONCEPTS

• **Computing methodologies → Neural networks**.

## KEYWORDS

Graph Neural Networks, Fairness, Pre-trained Graph Models

**ACM Reference Format:**
Anonymous Author(s). 2018. Endowing Pre-trained Graph Models with Provable Fairness. In *Proceedings of Make sure to enter the correct conference title from your rights confirmation emai (Conference acronym 'XX)*. ACM, New York, NY, USA, 11 pages. https://doi.org/XXXXXXX.XXXXXXX

## 1 INTRODUCTION

Graph Neural Networks (GNNs) [33, 41] have achieved significant success in analyzing graph-structured data, such as social

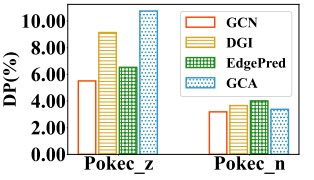 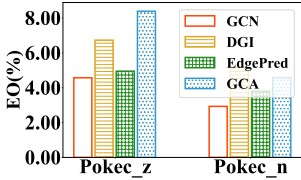

(a) Demographic Parity (DP)  (b) Equality Opportunity (EO)

**Figure 1: An example shows that testing the fairness of PGMs in downstream tasks. Fairness comparison of different methods is reported using DP (↓), EO (↓) on Pokec_z and Pokec_n datasets, using three pre-training methods (i.e., DGI, EdgePred, GCA) and vanilla GCN.**

network [12] or webpage network [39]. Recently, inspired by pre-trained language models, pre-trained graph models (PGMs) have received considerable attention in the field of graph machine learning. The aim is to capture transferable inherent structural properties and apply them to different downstream tasks [17, 36, 42]. As a powerful learning approach, PGMs have been applied in various domains such as social network analysis [31], recommendation systems [14], and drug discovery [37].

Due to the sensitive information such as gender, race, and religion present in pre-training corpora, it has been proven that pre-trained language models inherit biases from human society [44]. This naturally gives rise to the following questions: *Do pre-trained graph models also inherit bias on graphs?* In order to answer this question, we apply three different pre-training methods on datasets Pokec_z and Pokec_n. As depicted in Figure 1, we observe that PGMs inevitably capture sensitive attribute semantics during the pre-training phase, resulting in even more unfairness in downstream applications than vanilla GCN. Another question naturally arises: *How to improve the fairness of PGMs?* Since different downstream tasks may be associated with different sensitive attributes [5, 28, 40], debiasing during the pre-training phase may not be targeted to a specific task. Consequently, there is a need for a flexible approach to debiasing in downstream tasks.

In recent years, an increasing number of methods have been proposed for fair GNNs [6, 8, 21]. They design various fairness-related losses to constrain the parameter optimization process of GNNs. For example, methods based on counterfactual fairness [11, 26, 43] introduce similarity loss that aims to maximize the similarity between the original representations and counterfactual representations, to learn node embeddings that are invariant to sensitive attribute. Approaches based on sensitive attribute classifiers [6, 38] introduce adversarial loss to constrain GNNs from capturing sensitive information, or to generate new graphs with fair topology and node features [24].

Although the above methods achieve success for fair GNNs, when employed for addressing fairness issues of PGMs, they have two main drawbacks: (1) Their debiasing process is coupled with the

parameter optimization of GNNs. Due to that different downstream tasks may have different sensitive attributes, maintaining a specific PGM for each downstream task is not efficient. (2) Most existing fairness methods lack theoretical analysis and guarantees [2, 19], meaning that they do not provide a practical guarantee, i.e., provable lower bounds on the fairness of model prediction. This is important in practical scenarios when determining whether to deploy the models [4, 9, 18, 32].

To address the above limitations, we propose an adapter tuning framework called GraphPAR, which freezes the parameters of PGMs and applies a parameter-efficient adapter on node representations to make the model's predictions fairer. Specifically, we first design an augmenter that extends the node representation with different sensitive attribute semantics for each node via linear interpolation. Based on the augmented node representations, we utilize two adversarial debiasing techniques to optimize adapter parameters, preventing the propagation of sensitive attribute semantics from PGMs to downstream task predictions. Furthermore, with GraphPAR, we quantify whether the fairness of each node is provable, i.e., predictions are always fair within a certain range of sensitive attribute semantics. For example, when a person's gender semantics gradually transit from male to female, our provable fairness guarantees that the prediction results will not change. In summary, our debiasing framework GraphPAR is applicable to any PGMs while providing fairness with theoretical guarantees. Experimental evaluations conducted on three real-world datasets and three pre-training methods demonstrate that GraphPAR achieves state-of-the-art performance and fairness on node classification task. Moreover, with the help of GraphPAR, around 90% of nodes have provable fairness. The main contributions of this work can be summarized as follows:

(1) We explore fairness in PGMs for the first time and discover that PGMs may capture more sensitive attribute semantics than supervised GNNs during the pre-training phase, resulting in unfairness in downstream applications.

(2) We propose a novel approach to endow PGMs with fairness during the adaptation for downstream tasks. Specifically, we design GraphPAR, which utilizes a sensitive semantic augmenter and two adversarial debiasing methods to improve the fairness of PGMs while providing theoretical guarantees for fairness.

(3) We conduct extensive experiments on three real-world datasets to demonstrate the effectiveness of our model in achieving fair predictions and providing provable fairness.

## 2 RELATED WORK

### 2.1 Pre-trained Graph Models

In the pre-training phase, owing to its huge model parameters, PGMs can capture abundant knowledge from massive labeled and unlabeled graph data. Based on pre-training tasks, the existing pre-training methods mainly can be categorized into two categories: contrastive pre-training and predictive pre-training. Contrastive methods aim to maximize mutual information between different views, encouraging the model to capture invariant semantic information across various perspectives. For example, DGI [36] and InfoGraph [34] present approaches that aim to generate expressive representations for either graphs or nodes by maximizing

the mutual information between graph-level representations and substructure-level representations at various levels of granularity. GraphCL [46] and its variants [35, 45] introduce a range of sophisticated augmentation strategies for graph-level pre-training. Different from contrastive methods, predictive pre-training methods aim to equip graph models with an understanding of the universal structural and attribute semantics of graphs. For instance, attribute masking is proposed by [17] where the input node/edge attributes are randomly masked, and the GNN is asked to predict them. EdgePred [13] samples negative sample edges and utilizes a general GNN encoder to predict edge existence during the pre-training phase. GraphMAE [16] addresses the overemphasis on structure information in previous predictive methods by incorporating feature reconstruction and a re-mask decoding strategy for self-supervised learning.

Despite the ability of PGMs to capture abundant knowledge that proves valuable for downstream tasks, the conventional fine-tuning process still has some drawbacks, such as overfitting, catastrophic forgetting, and parameter inefficiency[23, 43]. To alleviate these issues, recent research has focused on developing parameter-efficient tuning (delta tuning) techniques that can effectively adapt pre-trained models to new tasks [7]. Delta tuning [7] seeks to tune a small portion of parameters and keep the left parameters frozen. For example, prompt tuning [25] aims to modify model inputs rather than model architecture. Adapter tuning [23] trains only a fraction of the Adapter's parameters to make PGMs adapt to downstream tasks.

Though a large number of research have been proposed on how to design pre-training methods and how to fine-tune PGMs in downstream tasks, most of them focus on how to improve performance while ignoring their plausibility in terms of fairness and so on.

### 2.2 Fairness of Graph

Recent studies [6, 21] show that GNNs tend to inherit bias from training data and the message-passing mechanism of GNNs and graph structure could magnify the bias [6]. Hence, many efforts have been made for fair GNNs, generally categorized into three different phases [2]. Pre-processing techniques remove bias or unfairness before GNN training occurs, by targeting the input graph structure, input features, or both. For instance, EDITS [8] propose a novel approach that utilizes the Wasserstein distance to address both attribute-based bias and structural bias in GNNs, effectively mitigating these biases. In-training techniques focus on modifying the objective function of GNNs to learn fair and unbiased embeddings during training. For example, NIFTY [1] propose a novel multiple objective function, which incorporates fairness and stability considerations. Graphair [24] introduces an automated augmentation model that generates new fair graphs to achieve fairness and informativeness simultaneously. A few post-processing methods have been proposed to remove bias from GNNs. FairGNN [6] designs a framework that leverages GNNs for fair node classification when only limited sensitive attribute information is available. FLIP [27] addresses the problem of link prediction homophily with postprocessing, as well as an adversarial framework.

Although all the aforementioned works have achieved significant success in graph fairness issues, most methods require optimizing

the parameters of the GNN. Therefore, these methods cannot efficiently address the fairness of PGMs in downstream tasks. In addition, they all lack theoretical analysis and fairness guarantees, which are important in determining whether to deploy a model in a real-world scenario.

## 3  PROBLEM DEFINITION

### 3.1  Notations

Given an attributed graph as $\mathcal{G} = (\mathcal{V}, \mathcal{E}, \mathbf{X})$, where $\mathcal{V} = \{v_1, ..., v_n\}$ represents the set of $n$ nodes, $\mathcal{E} \subseteq \mathcal{V} \times \mathcal{V}$ represents the set of edges, $\mathbf{X} = \{\mathbf{x}_1, \ldots, \mathbf{x}_n\}$ represents the node features and $\mathbf{x_i} \in \mathbb{R}^d$. The adjacency matrix of the graph $\mathcal{G}$ is denoted as $\mathbf{A} \in \mathbb{R}^{n \times n}$, where $\mathbf{A}_{ij} = 1$ if nodes $v_i$ and $v_j$ are connected, otherwise $\mathbf{A}_{ij} = 0$. Each node $i$ is associated with a binary sensitive attribute $s_i \in \{0, 1\}$ (we assume one single, binary sensitive attribute for simplicity, but our method can easily handle multivariate sensitive attributes as well). Furthermore, we consider a PGM or GNN denoted as $f$, which takes the graph structure and node features as input and produces node representations. The encoded representations for the $n$ nodes are denoted by $\mathbf{H} = \{\mathbf{h}_i\}_{i=1}^n$, where $\mathbf{H} = f(\mathcal{V}, \mathcal{E}, \mathbf{X})$ and $\mathbf{h}_i \in \mathbb{R}^p$.

In the pre-training phase, the parameters of a PGM $f$ is optimized via some self-supervised methods, such as graph contrastive learning [34, 36, 45, 46] or graph context prediction [13, 15–17]. In downstream tasks, adapter tuning by freezing the parameter $f_\theta$ of PGMs and just training the parameter $g_\theta$ of adapter $g$ to adapt PGMs for downstream tasks. Generally, $|g_\theta| \ll |f_\theta|$, where $|\cdot|$ denotes the number of parameter. In adapter, given an input $\mathbf{h}_i \in \mathbb{R}^p$, a down projection projects the input to a $q$-dimensional space, after which a nonlinear function is applied. Then the up-projection maps the $q$-dimensional representation back to $p$-dimensional space.

### 3.2  Fairness Definition on Graph

Fairness definition on graph refers to the prediction that the model should not be influenced by its sensitive attribute [26].

DEFINITION 1 (FAIRNESS ON GRAPH). *Given a graph* $\mathcal{G} = (\mathcal{V}, \mathcal{E}, \mathbf{X})$, *the encoder* $f(\cdot)$ *and the classifier* $d(\cdot)$ *trained on this graph satisfies fairness if for any node* $v_i$ :

$$P((d(f(\mathbf{X}, \mathbf{A})_i))_{S \leftarrow s} | \mathbf{X}, \mathbf{A}) = P((d(f(\mathbf{X}, \mathbf{A})_i))_{S \leftarrow s'} | \mathbf{X}, \mathbf{A}), \ s.t. \ \forall s \neq s',$$
(1)

*where* $s, s' \in \{0, 1\}^n$ *are two arbitrary sensitive attribute values. In other words, such a definition requires the prediction result will not change as the sensitive attribute value variations.*

### 3.3  Fairness Definition on PGMs

In order to improve the fairness of PGMs in downstream tasks, it is necessary to define fairness for PGMs. Combining the setting of PGMs and Definition 1, the fairness of PGMs in downstream tasks is defined in this work as follows:

DEFINITION 2 (FAIRNESS OF PGMS IN DOWNSTREAM TASKS). *A PGM* $f$ *is fair in the downstream tasks if the model predictions are the same under different sensitive attribute semantics, as formally defined below:*

$$P((d(g(h_i))_{\mathbf{S}_h \leftarrow \mathbf{s}}) = P((d(g(h_i))_{\mathbf{S}_h \leftarrow \mathbf{s}'}), \ s.t. \ \forall ||\mathbf{s} - \mathbf{s}'||_2 \neq 0, \ (2)$$

*where* $\mathbf{S}_h$ *is a vector with the same dimension as the representation* $\mathbf{h}$*, and its value represents the node's sensitive attribute semantics.* $\mathbf{s}$ *and* $\mathbf{s}'$ *are different semantics representations of sensitive attribute.*

The above fairness definition implies that prediction results should be the same as the sensitive attribute semantics representation variations. For example, when a person's gender semantics representation gradually transits from male to female, fairness is satisfied if the model predictions are always consistent, otherwise not satisfied.

## 4  METHODOLOGY

According to Definition 2, in order to improve fairness for PGMs in downstream tasks, in this section, we propose an adapter-tuning framework called GraphPAR, which parameter-efficient improves the fairness of PGMs. Adapter tuning is illustrated in Figure 2 (a) and consists of two components: (1) Sensitive semantic augmenter. We first compute a vector $\boldsymbol{\alpha}$ about sensitive attribute semantics, and then extend the node representation with different sensitive attribute semantics for each node via linear interpolation on $\boldsymbol{\alpha}$. (2) Training adapter. We employ random augmentation adversarial debiasing and Min-max adversarial attack debiasing to tune the parameters of the adapter.

### 4.1  Sensitive Semantic Augmenter

In order to extend the node representation with different sensitive attribute semantics for each node, we design a sensitive semantic augmenter. Initially, we leverage the known sensitive attribute information and representations of the nodes to calculate a sensitive attribute semantics vector $\boldsymbol{\alpha}$. Subsequently, we extend the node representation $\mathbf{h}_i$ for each node via linearly interpolating in the direction of $\boldsymbol{\alpha}$, obtaining the sensitive attribute semantics augmentation set $\mathcal{S}_i$.

**Computing the sensitive attribute semantics vector** $\boldsymbol{\alpha}$**.** Leveraging the capabilities of PGMs in capturing both graph structure and node attributes, we expect to derive a vector $\boldsymbol{\alpha}$ that effectively represents the sensitive attribute semantics. Firstly, we utilize the given PGM $f$ to obtain node representations $\mathbf{H}$. Then, based on known nodes' sensitive attribute $s$, we partition the node representations into positive and negative sets, i.e., $\mathbf{H}_{pos}$ and $\mathbf{H}_{neg}$. We calculate the average representation $\mathbf{h}_{pos}$ for nodes possessing the sensitive attribute and $\mathbf{h}_{neg}$ for nodes lacking the sensitive attribute, both obtained from $\mathbf{H}_{pos}$ and $\mathbf{H}_{neg}$, respectively. Lastly, the difference between $\mathbf{h}_{pos}$ and $\mathbf{h}_{neg}$ represents the sensitive attribute semantics vector:

$$\boldsymbol{\alpha} = \mathbf{h_{pos}} - \mathbf{h_{neg}}, \tag{3}$$

$$\mathbf{h}_{pos} = \frac{1}{n_{pos}} \sum_{i=1}^{n_{pos}} \mathbf{H}_{pos,i}, \mathbf{h}_{neg} = \frac{1}{n_{neg}} \sum_{i=1}^{n_{neg}} \mathbf{H}_{neg,i}, \tag{4}$$

where $n_{pos}$ and $n_{neg}$ denote the number of positive and negative samples.

With the vector $\boldsymbol{\alpha}$, we expect to move in the direction of $\boldsymbol{\alpha}$ to increase the presence of the sensitive attribute, while moving in the opposite direction diminishes its presence. Thus, to verify whether $\boldsymbol{\alpha}$ satisfies our expectations, we conduct a test experiment using $\boldsymbol{\alpha}$. First, we divide $\mathbf{H}$ into training and test sets. In the training

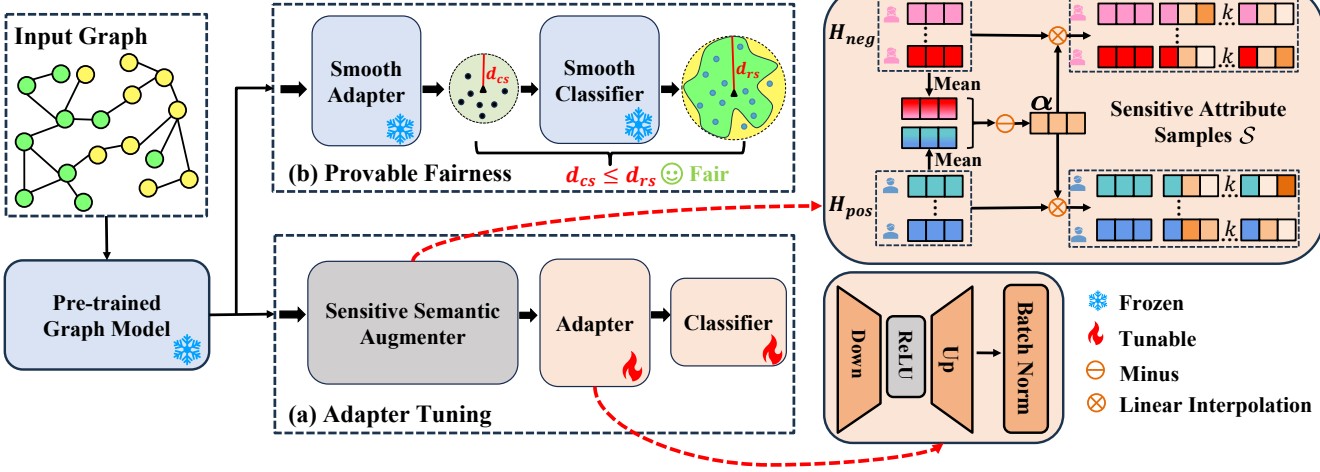

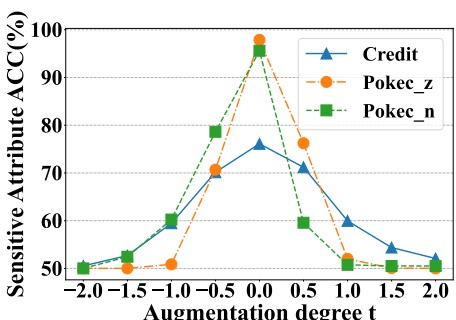

**Figure 2: Overview of GraphPAR. In the adapter tuning phase, we first utilize the PGMs to obtain node representations H. Then, we design a sensitive semantic augmenter to augment the node representations with different sensitive attribute semantics, i.e., sensitive attribute samples $\mathcal{S}$. Finally, we use the augmented node representations to train an adapter, improving the fairness of PGMs. In the provable fairness phase, based on the smoothed versions of the trained adapter and classifier, we use the smooth adapter to get its output bound guarantee $d_{cs}$, and use the smooth classifier to get its local robustness guarantee $d_{rs}$. Sequentially, we quantify whether the fairness of each node is provable by comparing $d_{cs}$ with $d_{rs}$.**

set, we train a sensitive attribute classifier $d_{sens}$ and compute the $\boldsymbol{\alpha}$. Next, we move the node representations in the test set along the direction of $\boldsymbol{\alpha}$ with varying augmentation degree $t$. Lastly, we utilize the trained classifier $d_{sens}$ to predict the accuracy of the sensitive attribute on the test set. The results are presented in Figure 3, revealing the following findings:

(1) When no movement is performed, i.e., $t = 0$, the prediction accuracy of the sensitive attribute is highest. This indicates that the pre-training may inevitably capture the sensitive attribute information present in the dataset.

(2) As the magnitude of the augmentation degree $|t|$ increases, the prediction accuracy of $d_{sens}$ gradually decreases until it reaches 50%. This is because modifying the node representations along the same sensitive semantics direction leads all nodes to become increasingly similar in terms of sensitive attribute semantics. For instance, when moving in the direction of $t > 0$, nodes initially classified as negative samples shift to positive samples, while nodes previously classified as positive samples become more strongly associated with the sensitive attribute. Consequently, the sensitive attribute classifier $d_{sens}$ can only accurately classify half of the nodes.

**Augmenting sensitive attribute semantics for h.** After verifying the effectiveness of $\boldsymbol{\alpha}$, we employ it to augment a set of sensitive attribute $\mathcal{S}_i$ for each node representation $\mathbf{h}_i$. This augmentation is achieved through a linear interpolation method and can be expressed as:

$$\mathcal{S}_i := \{\mathbf{h}_i + t \cdot \boldsymbol{\alpha} \mid |t| \leq \epsilon\} \subseteq \mathbb{R}^p, \quad (5)$$

Here, $\epsilon$ represents the augmentation range applied to the direction of the sensitive attribute semantics. The above augmentation method offers two key advantages: (1) It efficiently extends node

**Figure 3: Under different augmentation degree $t$ of sensitive attribute semantics, comparing sensitive attribute prediction accuracy against different datasets.**

representations with different semantics of sensitive attributes as line segments. These line segments correspond to multiple points in the original input space, thereby bypassing complex augmentation designs in the original input space [43, 47]. (2) Although this work primarily focuses on single sensitive attribute scenarios, this method can be simply extended to situations involving multiple sensitive attributes. In such cases, interpolation can be performed along multiple sensitive attribute semantics vectors.

## 4.2 Training Adapter for PGMs Fairness

Given any PGMs $f$ and the sensitive attribute augmentation set $\mathcal{S}$, we will now outline how to improve the fairness of PGMs by training a parameter-efficient adapter $g$, while also ensuring the performance of downstream tasks. Specifically, we employ two

debiasing methods for training the adapter: random augmentation adversarial training and min-max adversarial attacks training.

**Random augmentation adversarial training (RandAT).** During the adapter $g$ training process, we adopt a strategy where we choose $k$ samples from the augmented sensitive attribute set $\mathcal{S}_i$ to obtain adversarial training set $\hat{\mathcal{S}}_i$, i.e.,

$$\hat{\mathcal{S}}_i = \{\mathbf{h}_i + t_j \cdot \boldsymbol{\alpha}\}_{j=1}^{k}, \; t_j \sim \text{Uniform}(-\epsilon, \epsilon), \qquad (6)$$

where $\epsilon$ represents the augmentation range. These selected samples are then incorporated into the training of the adapter. The optimization loss can be formulated as:

$$\mathcal{L}_{\text{RandAT}} = \mathbb{E}_{i \in \mathcal{V}_L} \left[ \mathbb{E}_{\mathbf{h}' \in \{\mathbf{h}_i\} \cup \hat{\mathcal{S}}_i} \left[ \ell(d \circ f(\mathbf{h}'), y_i) \right] \right], \qquad (7)$$

where $\mathcal{V}_L$ is the set of labeled nodes, $d$ is a downstream classifier, and $\ell(\cdot)$ is cross-entropy loss which measures the prediction error.

In RandAT, by including a diverse range of sensitive attribute semantic samples in the tuning process, both the adapter $g$ and the classifier $d$ become capable of handling variations in sensitive information. This enables them to generalize effectively to samples with different sensitive attribute semantics. Consequently, this approach helps mitigate potential discriminatory predictions, as the adapter becomes more robust and adaptable to changes in sensitive attribute semantics.

**Min-max adversarial attacks training (MinMax).** Unlike RandAT, the thought behind MinMax is to find and optimize a worst-case scenario in each round. Our objective is to minimize the discrepancy between the representation $\mathbf{h}_i$ and its corresponding augmented sensitive attribute semantics set $\mathcal{S}_i$. This is achieved by ensuring that the representation $\mathbf{h}_i$ closely aligns with the representations within $\mathcal{S}_i$. To quantify this alignment, we seek to minimize the distance between $\mathbf{h}_i$ and $\mathcal{S}_i$. Hence, our optimization objective entails minimizing the following loss function:

$$\mathcal{L}_{MinMax}(\mathbf{h}_i) = \max_{\mathbf{h}'_i \in \mathcal{S}_i} \left\| g(\mathbf{h}_i) - g(\mathbf{h}'_i) \right\|_2, \qquad (8)$$

where $\mathbf{h}'_i$ is the any sample of $\mathcal{S}_i$. Minimizing $\mathcal{L}_{adv}(\mathbf{h}_i)$ is a min-max optimization problem, and adversarial training is effective in this scenario. Since the input domain of the inner maximization problem is a simple line segment about $\boldsymbol{\alpha}$, we can perform adversarial training [10] by uniformly sampling $k$ points from $\mathcal{S}_i$ to construct $\hat{\mathcal{S}}_i$ and approximate it as follow:

$$\mathcal{L}_{MinMax}(\mathbf{h}_i) \approx \max_{\mathbf{h}'_i \in \hat{\mathcal{S}}_i} \left\| g(\mathbf{h}_i) - g(\mathbf{h}'_i) \right\|_2, \qquad (9)$$

where $\mathbf{h}'_i$ have different sensitive attribute semantics with $\mathbf{h}_i$. To ensure that the adapter $g$ does not filter out useful task information, we introduce a downstream task classifier after the adapter. We add an additional cross-entropy classification loss term to ensure the performance of the downstream task:

$$\mathcal{L}_{cls}(\mathbf{h}_i, y_i) = \ell(d \circ f(\mathbf{h}_i), y_i). \qquad (10)$$

The final optimization objective is as follows:

$$\mathcal{L} = \lambda \mathcal{L}_{MinMax} + \mathcal{L}_{cls}, \qquad (11)$$

where $\lambda$ is a scale factor with respect to the fairness loss, which is used to balance accuracy and fairness.

## 5 PROVABLE FAIR ADAPTATION OF PGMS

In this section, based on GraphPAR, we will primarily discuss how to provide provable fairness for each node, i.e., the prediction results are consistent within a certain range of sensitive attribute semantics. We divide this process into two key components as depicted in Figure 2 (b): (1) Smooth adapter. We construct a smoothed version for the adapter using center smooth, which provides a bound for the output variation of node representation $\mathbf{h}_i$ within the range of sensitive attribute semantics change. This guarantees that the range of output results is contained within a minimal enclosing ball centered at $\mathbf{z}$ with a radius of $d_{cs}$. (2) Smooth classifier. We construct a smoothed version for the classifier using random smooth, which provides a local robustness against the center $\mathbf{z}$. By determining whether all points within the minimum enclosing ball are classified into the same class, i.e., $d_{cs} < d_{rs}$, we quantify if the fairness of each node is provable.

### 5.1 Provable Adaptation

To guarantee the range of change in the representation after applying the adapter $g$, we employ a technique called center smoothing [22] to obtain a smoothed version of the adapter, denoted as $\hat{g}$. It provides a guarantee for the output bound of the adapter with a representation $\mathbf{t}$ as the input, described in Theorem 1:

THEOREM 1 (CENTER SMOOTHING [22]). *Let $\hat{g}$ denote an approximation of the smoothed version of the adapter $g$, which maps a representation $\mathbf{h}$ to the center point $\hat{g}(\mathbf{h})$ of a minimum enclosing ball containing at least half of the points $\mathbf{z} \sim g(\mathbf{h} + \mathcal{N}(0, \sigma_{cs}^2 I))$. Then, for an $l_2$-perturbation size $\epsilon_1 > 0$ on $\mathbf{h}$, we can produce a guarantee $d_{cs}$ of the output change with confidence $1 - \alpha_{cs}$:*

$$\forall \mathbf{h}' \; s.t. \|\mathbf{h} - \mathbf{h}'\|_2 \le \epsilon_1, \|\hat{g}(\mathbf{h}) - \hat{g}(\mathbf{h}')\|_2 \le d_{cs}. \qquad (12)$$

In adapter, since $\mathbf{h}' - \mathbf{h} = t \cdot \boldsymbol{\alpha}$, we have $\epsilon_1 = t\|\boldsymbol{\alpha}\|_2$. Theorem 1 implies that given a node representation $\mathbf{h}$ and its set of sensitive attribute samples $\mathcal{S}_i$, a guarantee $d_{cs}$ can be computed with high probability. This $d_{cs}$ represents the range of adapter's output changes, serving as a meaningful certificate. It guarantees that when the sensitive attribute semantics of input $\mathbf{h}$ is perturbed within a range defined by $\epsilon$, the range of output remains within a minimal enclosing ball.

### 5.2 Provable Classification

Subsequently, it is necessary to demonstrate that predictions for all points within this minimum enclosing ball are classified consistently. This consistency guarantees the effectiveness of debiasing results.

THEOREM 2. *(Random Smoothing [3]) Let $d$ be a classifier and let $\varepsilon \sim \mathcal{N}(0, \sigma_{rs}^2 I)$. The smoothing version of the classifier $\hat{d}$ is defined as follows:*

$$\hat{d}(\mathbf{z}) = \arg \max_{y_i \in \mathcal{Y}} \mathbb{P}_{\varepsilon}(d(\mathbf{z} + \varepsilon) = y_i). \qquad (13)$$

*Suppose $y_A \in \mathcal{Y}$ and $\underline{p_A}, \overline{p_B} \in [0, 1]$ satisfy:*

$$\mathbb{P}_{\varepsilon}(d(\mathbf{z} + \varepsilon) = y_A) \ge \underline{p_A} \ge \overline{p_B} \ge \max_{y_B \ne y_A} \mathbb{P}_{\varepsilon}(d(\mathbf{z} + \varepsilon) = y_B). \qquad (14)$$

Then, we have $\widehat{d}(z + \delta) = y_A$ for all $\delta$ satisfying $\|\delta\|_2 < d_{rs}$, where $d_{rs}$ can be obtain as follow:

$$d_{rs} := \frac{\sigma_{rs}}{2}(\Phi^{-1}(\underline{p_A}) - (\Phi^{-1}(\overline{p_B})), \tag{15}$$

where $\mathcal{Y}$ denotes the set of class labels, $\Phi$ is the cumulative distribution function (CDF) of the standard normal distribution $\mathcal{N}(0, 1)$, and $\Phi^{-1}$ is its inverse.

Theorem 2 derives a local robustness radius $d_{rs}$ for the classifier's input by employing the smoothed version $\widehat{d}$ of the classifier $d$. This robustness guarantees that within the verified region of input, which is bounded by $d_{rs}$, the classification output of $\widehat{d}$ remains unchanged, providing a guarantee of stability and consistency in the classifier's predictions. Theorem 2 is especially important for providing provable fairness, because if $d_{cs} < d_{rs}$, then it guarantees consistency in the predictions to different sensitive attribute semantic samples.

## 5.3 GraphPAR Provides Provable Fairness

To establish a theoretical guarantee for the debiasing effect of adapter $g$ and verify the fairness of the debiasing process, we define the provable fairness of PGMs as follows:

Definition 3 (Provable Fairness of PGMs in downstream tasks). *Given a node representation* $\mathbf{h}$*, the debiasing process $M$ satisfies:*

$$M(\mathbf{h}) = M(\mathbf{h}'), \forall h' \in \mathcal{S}, \tag{16}$$

*where $\mathcal{S}$ is the set of sensitive attribute augmentations for* $\mathbf{h}$*.*

With the utilization of the aforementioned two smoothing techniques, the provable fairness of PGMs is naturally achieved with the following theorem:

Theorem 3. *Assuming we have a PGM $f$, a center smoothing adapter $\widehat{g}$, and a random smoothing classifier $\widehat{d}$. For node $i$, if $\widehat{g}$ obtains a output guarantee $d_{cs}$ with confidence $1 - \alpha_{cs}$ and $\widehat{d}$ obtains a local robustness guarantee $d_{rs}$ with confidence $1 - \alpha_{rs}$, and satisfy $d_{cs} < d_{rs}$, then the fairness of the debiasing $M = \widehat{d} \circ \widehat{g} \circ f(\mathcal{V}, \mathcal{E}, \mathbf{X})_i$ is provable with a confidence $1 - \alpha_{cs} - \alpha_{rs}$.*

Proof. Assume that Theorem 3 holds for the node $i$. We need to show that with probability at least $1 - \alpha_{cs} - \alpha_{rs}$:

$$\forall h' \in S : \widehat{d} \circ \widehat{g}(\mathbf{h}) = \widehat{d} \circ \widehat{g}(h'), \tag{17}$$

where $\mathbf{h} \in \mathbf{H}, \mathbf{H} = f(\mathcal{V}, \mathcal{E}, \mathbf{X})$.

Next, recall the definition of $g_h(t) := g(\mathbf{h} + t \cdot \boldsymbol{\alpha})$ and note that for $\mathbf{h}' = \mathbf{h} + t' \cdot \boldsymbol{\alpha}$, the center smoothing of

$$\widehat{g_{h'}}(t) \sim g_{\mathbf{h}'}(t + \mathcal{N}(0, \sigma_{cs}^2)) = g(\mathbf{h}' + (t + \mathcal{N}(0, \sigma_{cs}^2)) \cdot \boldsymbol{\alpha}),$$

$$\widehat{g_h}(t + t') \sim g_{\mathbf{h}}(t + t' + \mathcal{N}(0, \sigma_{cs}^2)) = g(\mathbf{h} + (t + t' + \mathcal{N}(0, \sigma_{cs}^2)) \cdot \boldsymbol{\alpha}).$$

Since $\mathbf{h}' = \mathbf{h} + t' \cdot \boldsymbol{\alpha}$, the sampling distributions are the same, hence $\widehat{g_{h'}}(t) = \widehat{g_h}(t + t')$, and in particular $\widehat{g}(\mathbf{h}') = \widehat{g_{h'}}(0) = \widehat{g_h}(t')$

Now, let us get back to Equation 17. By definition of $\mathcal{S}$, for all $\mathbf{h}' \in \mathcal{S}, \mathbf{h}' = \mathbf{h} + t' \cdot \boldsymbol{\alpha}$ for some $t' \in [-\epsilon, \epsilon]$. Moreover, $\mathbf{z_{cs}} = \widehat{g}(\mathbf{h}) = \widehat{g_h}(0)$ and $\widehat{g}(\mathbf{h}') = \widehat{g_h}(t')$. Theorem 1 tells us that with confidence $1 - \alpha_{cs}$:

$$\left\|\widehat{g_h}(0) - \widehat{g_h}(t')\right\|_2 \leq d_{cs}, \forall t' \in [-\epsilon, \epsilon]$$

$$\Longleftrightarrow \left\|\mathbf{z_{cs}} - \widehat{g}(\mathbf{h}')\right\|_2 \leq d_{cs}, \forall \mathbf{h}' \in \mathcal{S}, \tag{18}$$

### Table 1: Datasets Statistics.

| Dataset | Credit | Pokec_n | Pokec_z |
|---|---|---|---|
| #Nodes | 30,000 | 66,569 | 67,797 |
| #Features | 13 | 266 | 277 |
| #Edges | 1,436,858 | 729,129 | 882,765 |
| Node label | Future default | Working field | Working field |
| Sensitive attribute | Age | Region | Region |
| Avg. degree | 95.79 | 16.53 | 19.23 |

provided that the center smoothing computation of $d_{cs}$ does not abstain.

Finally, we consider the last component of the pipeline – the smoothed classifier $\widehat{d}$. Provided that $\widehat{d}$ does not abstain at the input $d_{cs}$, Theorem 2 provides us with a radius $d_{rs}$ around $z_{cs}$ such that with confidence $1 - \alpha_{rs}$:

$$\widehat{d}(\mathbf{z_{cs}}) = \widehat{d}(\mathbf{z_{cs}} + \delta), \forall \delta \text{ s.t. } \|\delta\|_2 < d_{rs}$$

$$\Longleftrightarrow \widehat{d}(\mathbf{z_{cs}}) = \widehat{d}(\mathbf{z}'), \forall \mathbf{z}' \text{ s.t. } \left\|\mathbf{z_{cs}} - \mathbf{z}'\right\|_2 < d_{rs}. \tag{19}$$

If $d_{cs} < d_{rs}$, combining the conclusions in Equation 18 and Equation 19 and applying the union bound, we obtain that with confidence $1 - \alpha_{cs} - \alpha_{rs}$ we have $\widehat{d}(\mathbf{r_{cs}}) = \widehat{d}(\widehat{g}(\mathbf{h}'))$ for all $\mathbf{h}' \in S$, that is,

$$\forall \mathbf{h}' \in S(\mathbf{h}) : \widehat{d} \circ \widehat{g}(\mathbf{h}) = \widehat{d} \circ \widehat{g}(\mathbf{h}') \tag{20}$$

as required by Definition 3. The same proof technique can also be extended to the multiple sensitive attribute vectors case. □

The detailed algorithms process of Theorem 3 is referred to Appendix 1.

## 6 EXPERIMENTS

In this section, we extensively evaluate GraphPAR to answer the following research questions (RQs):

- **RQ1**: How effective is GraphPAR compared to existing graph fairness methods?
- **RQ2**: Compared to methods without debiasing adaptation, does GraphPAR show improvement in the number of nodes with provable fairness?
- **RQ3**: How do different hyperparameters of GraphPAR impact the classification performance and fairness?
- **RQ4**: How parameter-efficient is GraphPAR?

**Datasets.** These are common graph datasets with sensitive attributes collected from various domains, we choose three public datasets *Credit* [43], *Pokec_z* and *Pokec_n* [6]. As for *Credit*, Credit encompasses a network of individuals who are connected due to the likeness of their spending and payment habits. The sensitive attribute is the age of these individuals, and the objective is to predict whether their default payment method is credit card or not. As for *Pokec_z* and *Pokec_n*, datasets are created by sampling from Pokec based on geographic regions. Pokec encompasses anonymized data from the complete social network in 2012, encompassing user profiles that include details like gender, age, hobbies, interests, education, occupation, and so forth. The sensitive attribute is region and the working field is used as the predicted label. Detailed statistics are listed in Table 1.

                                                          

**Table 2: Performance and fairness ($\% \pm \sigma$) on node classification. The best results are in bold and runner-up results are underlined.**

| Method | | Credit | | | | Pokec_z | | | | Pokec_n | | | |
|---|---|---|---|---|---|---|---|---|---|---|---|---|---|
| | | ACC (↑) | F1 (↑) | DP (↓) | EO (↓) | ACC (↑) | F1 (↑) | DP (↓) | EO (↓) | ACC (↑) | F1 (↑) | DP (↓) | EO (↓) |
| | GCN | 69.73±0.04 | 79.14±0.02 | 13.28±0.15 | 12.66±0.24 | 67.54±0.48 | 68.93±0.39 | 5.51±0.67 | 4.57±0.29 | **70.11±0.34** | 67.37±0.38 | 3.19±0.86 | 2.93±0.95 |
| | FairGNN | 72.50±4.09 | 81.80±3.86 | 9.20±3.35 | 7.64±3.58 | 67.47±1.12 | 69.35±3.14 | 1.91±1.01 | 1.04±1.11 | 68.42±2.04 | 64.34±2.32 | 1.41±1.30 | 1.50±1.23 |
| | NIFTY | 70.89±0.59 | 80.23±0.54 | 9.93±0.59 | 8.79±0.71 | 65.83±3.90 | 66.99±4.26 | 5.47±2.13 | 2.64±1.02 | 68.97±1.21 | 66.77±1.27 | 1.68±0.90 | 1.38±0.91 |
| | EDITS | 66.80±1.03 | 76.64±1.13 | 10.21±1.14 | 8.78±1.15 | OOM | OOM | OOM | OOM | OOM | OOM | OOM | OOM |
| DGI | Naive | 75.72±2.18 | 84.73±2.00 | 7.87±2.22 | 6.51±2.79 | 67.87±0.51 | 70.23±0.80 | 4.69±1.95 | 3.03±1.34 | 68.58±1.22 | 65.66±1.37 | 3.58±3.09 | 4.99±3.68 |
| | GraphPAR$_{RandAT}$ | **76.88±1.33** | **85.85±1.36** | 5.93±2.91 | 4.44±3.34 | 67.05±1.33 | **70.50±0.69** | 1.90±1.22 | 0.84±0.28 | 68.92±1.55 | 65.61±1.33 | **1.19±0.65** | 2.11±1.60 |
| | GraphPAR$_{MinMax}$ | 74.37±2.91 | 83.46±2.64 | **3.81±2.37** | **2.60±2.48** | **68.32±0.55** | 68.35±2.38 | 1.64±0.78 | **0.53±0.39** | 68.43±0.55 | **68.20±2.22** | 1.73±0.76 | 1.11±0.88 |
| EdgePred | Naive | 69.66±1.74 | 79.30±1.63 | 7.89±2.28 | 6.67±2.42 | 67.33±0.44 | 69.17±0.52 | 6.00±3.04 | 3.95±2.52 | 68.60±0.53 | 65.56±0.79 | 2.48±0.86 | 5.29±2.71 |
| | GraphPAR$_{RandAT}$ | 69.97±2.35 | 79.55±2.24 | 6.36±2.19 | 4.83±2.70 | 66.87±1.12 | 68.86±0.46 | 1.99±1.12 | 2.27±1.23 | 68.49±1.41 | 65.45±1.02 | 1.79±0.85 | 3.69±0.68 |
| | GraphPAR$_{MinMax}$ | 68.53±1.23 | 78.19±1.14 | 5.10±2.31 | 4.52±2.17 | 67.51±0.55 | 69.03±0.82 | **1.45±1.40** | 1.15±0.85 | 69.10±0.91 | 65.00±1.10 | 1.28±0.97 | 3.31±2.06 |
| GCA | Naive | 75.28±0.51 | 84.35±0.47 | 8.56±0.97 | 6.21±0.90 | 67.63±0.44 | 70.24±0.98 | 7.68±2.19 | 4.82±1.43 | 67.85±1.23 | 65.81±1.35 | 2.90±2.61 | 3.23±1.05 |
| | GraphPAR$_{RandAT}$ | 75.50±1.29 | 84.66±1.27 | 5.51±2.44 | 3.98±1.96 | 66.73±2.22 | 70.32±0.73 | 4.23±2.50 | 2.94±1.84 | 68.11±0.44 | 64.43±1.05 | 2.35±1.12 | 2.42±1.62 |
| | GraphPAR$_{MinMax}$ | 73.74±2.01 | 82.96±1.74 | 4.90±1.90 | 2.96±1.66 | 66.59±1.28 | 68.74±1.17 | 2.33±2.28 | 2.42±1.72 | 68.11±0.70 | 65.49±1.57 | 1.41±0.86 | **0.94±0.59** |

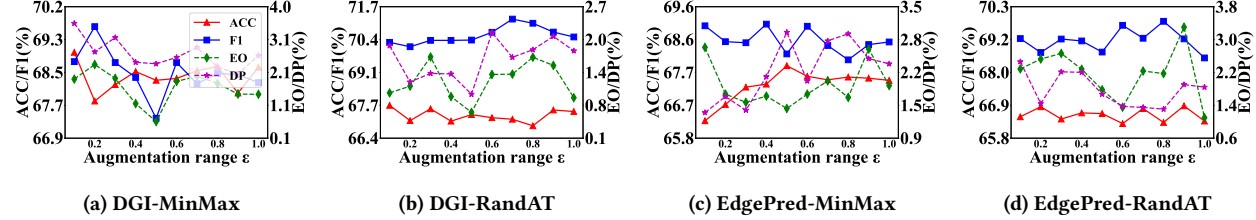

(a) DGI-MinMax     (b) DGI-RandAT     (c) EdgePred-MinMax     (d) EdgePred-RandAT

**Figure 4: The effect of augmentation range $\epsilon$ to GraphPAR$_{minmax}$ and GraphPAR$_{RandAT}$ in the Pokec_z dataset.**

**Baselines.** We compare GraphPAR to four baseline models: GCN [20] is the most common GNN; FairGNN [6] is a framework for fair node classification using GNNs given limited sensitive attribute information; NIFTY [43] achieves fairness by maximizing the similarity of representations learned from the original graph and their augmented counterfactual graphs. EDITS [8] debiases the input network to remove the sensitive information in the graph data. Since GraphPAR is based on PGMs, we include three types of PGM as baseline models: contrastive pretraining models DGI [36] and GCA [48] that maximize mutual information between different views, as well as predictive pretraining model EdgePred [13] that reconstructs masked edges as its task.

**Experiment setup.** For Credit we follow train/valid/test split in [43], and for Pokec_z and Pokec_n we follow in [6]. Unless otherwise specified, we set the hyperparameters as follows: For the sensitive semantics augmented, sensitive attribute semantics augmentation range $\epsilon = 0.5$, number of randomly selected augmentation samples $k = 20$, fairness loss scale factor $\lambda = 0.1$. For the adapter, the dimension size of down projection is half of the input dimension size, the learning rate is 0.01, and the number of epochs is 1000. We use GCN as the backbone for all PGMs, take the Adam optimizer and implement GraphPAR with Pytorch [29]. To provide further provable fairness, following the parameter settings in center smooth [22] and random smooth [3], we utilize the trained adapter and the classifier to obtain their smoothed versions respectively. We report the experiment results over five runs with different random seeds. The code and datasets will be publicly available after the review.

## 6.1 Prediction Performance and Fairness (RQ1)

We test GraphPAR on the node classification task. To evaluate the performance on classification and its fairness, we choose four metrics: accuracy (ACC) and macro-F1 (F1) score, measure how well the nodes are classified; demographic parity (DP) and equality of opportunity (EO), measure how fair the classification is. The results are shown in Table 2. The best results are shown in bold, while the runner-up results are underlined. We interpret the results as follows:

**GraphPAR outperforms baseline models both in classification and fairness performance.** GraphPAR are demonstrated to be superior in both classification and fairness performances, enhancing existing PGM models as well as outperforming other graph fairness methods. This result supports the effectiveness of GraphPAR to addresses fairness issues in the embedding space:

(1) Powerful pre-training strategies enable the embeddings to include intrinsic information for downstream tasks;

(2) Since PGMs also capture sensitive attribute information, the sensitive semantics vector can be effectively constructed;

(3) Augmenting in the embedding space is independent of task labels, thus the sensitive semantic augmenter does not corrupt the downstream performance.

**Performance of GraphPAR varies among different PGMs.** The performance of classification and fairness varies when choosing different PGMs as the backbone of GraphPAR. Usually, we observe that contrastive pre-training methods DGI and GCA perform better than the predictive method EdgePred, implying the importance of which PGM we are based on.

813
814
815
816
817
818
819
820
821
822
823
824
825
826
827
828
829
830
831
832
833
834
835
836
837
838
839
840
841
842
843
844
845
846
847
848
849
850
851
852
853
854
855
856
857
858
859
860
861
862
863
864
865
866
867
868
869
870

**Table 3: Provable fairness results for different training schemes. The best result on each metric is shown in bold.**

| Dataset | PGM | Naive | | GraphPAR$_{RandAT}$ | | GraphPAR$_{MinMax}$ | |
|---|---|---|---|---|---|---|---|
| | | ACC (↑) | Prov_Fair (↑) | ACC (↑) | Prov_Fair (↑) | ACC (↑) | Prov_Fair (↑) |
| Credit | DGI | 72.80 | 27.63 | **75.39** | 37.05 | 72.71 | **89.59** |
| | EdgePred | 66.87 | 5.41 | **67.02** | 44.20 | 66.41 | **96.28** |
| | GCA | 72.86 | 0.28 | **73.25** | 20.26 | 70.10 | **92.92** |
| Pokec_z | DGI | **67.30** | 1.47 | 67.21 | 10.99 | 67.28 | **94.51** |
| | EdgePred | 66.02 | 0 | 66.27 | 37.51 | **66.80** | **90.97** |
| | GCA | **66.92** | 13.9 | 66.67 | 16.14 | 65.22 | **95.75** |
| Pokec_n | DGI | **68.45** | 0.70 | 67.52 | 0.52 | 68.38 | **77.97** |
| | EdgePred | 67.58 | 0 | **68.15** | 21.17 | **68.15** | **88.76** |
| | GCA | 67.49 | 17.80 | **67.52** | 10.03 | 67.30 | **91.16** |

**RandAT and MinMax perform well, but in different ways.** It is worth mentioning that RandAT often achieves the best result on classification while MinMax often performs the best on fairness. The following differences in the training schemes directly lead to the result above:

(1) RandAT uses all augmented samples in downstream tasks while MinMax only uses the original data. As a result, RandAT often outperforms MinMax on classification metrics ACC and F1, regarding that classification benefits from data augmentation [30].

(2) To debias sensitive information, MinMax minimizes the largest distance between an individual $\mathbf{h}_i$ and other samples $\mathbf{h}_i'$ in the sensitive augmentation set $\mathcal{S}_i$, which can achieve a better debiasing result against the sampling strategy in RandAT that performs adversarial training on all augmented samples.

These empirical findings straightforwardly demonstrate the characteristics of RandAT in Equation 7 and MinMax in Equation 8.

## 6.2 Debiasing Guarantee (RQ2)

To additionally guarantee how fair the classification is, we evaluate the provable fairness of GraphPAR compared with naive PGMs. Here, the metrics are accuracy (ACC) and provable fairness (Prov_Fair) in Definition 3. The result is presented in Table 3. We have the following observations:

(1) Different from naive PGMs that show little or nearly zero provable fairness, RandAT achieves much better provable fairness, and MinMax has its fairness guaranteed very well. According to Theorem 3 where the provable fairness of PGMs satisfies $d_{cs} < d_{rs}$, since $d_{rs}$ is the same, but $d_{cs}$ is different among training schemes: naive PGMs do not optimize $d_{cs}$, thus the fairness is nearly not guaranteed; RandAT is trained with many samples with sensitive semantics augmented, which has a positive effect on minimizing $d_{cs}$ but not in an explicit way; MinMax achieves the best provable fairness by directly optimizing $d_{cs}$ with min-max training.

(2) Also, the classification performances of RandAT and MinMax are competitive to naive PGMs. RandAT does not lose its classification performance because its augmentation is performed in sensitive semantics and do not introduce noise to task-related information; on the other hand, MinMax trains the downstream classifier with original data after an adapter, implying that the adapter almost has no negative effect on the classification while guaranteeing fairness.

871
872
873
874
875
876
877
878
879
880
881
882
883
884
885
886
887
888
889
890
891
892
893
894
895
896
897
898
899
900
901
902
903
904
905
906
907
908
909
910
911
912
913
914
915
916
917
918
919
920
921
922
923
924
925
926
927
928

In conclusion, the empirical results above support that when trained with RandAT and MinMax, GraphPAR guarantees fairness without compromising its classification performance.

## 6.3 Hyperparameter Sensitivity Analysis (RQ3)

To further validate how the hyperparameters impact the performance of GraphPAR, we conduct sensitivity analysis experiments on the augmentation range $\epsilon$, augmentation sample number $k$, and fairness loss scale $\lambda$. As shown in Figure 4, the best $t, k, \lambda$ for fairness metrics varies among different PGMs, different datasets, and different training methods (RandAT and MinMax), but they consistently outperform naive PGMs, illustrating the effectiveness of the proposed GraphPAR. For example, on Pokec_z trained with MinMax, GraphPAR on DGI achieves the best fairness when $t = 0.5$, while $t = 0.3$ for EdgePred. For DGI trained with MinMax, GraphPAR achieves the best fairness on Pokec_z when $\lambda = 0.6$, while $\lambda = 0.7$ for Credit. A key observation is when $\epsilon$ is tuned between 0 and 1, ACC and F1 tend to be stable, while EO and DP fluctuate. This suggests that the sensitive semantic augmenter does not corrupt task-related information while successfully capturing sensitive information. We show more detailed results in the Appendix B.

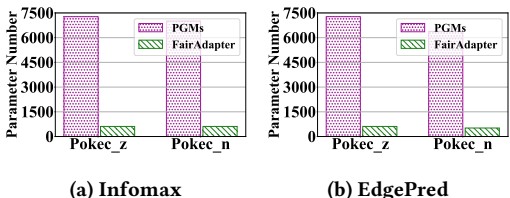

(a) Infomax          (b) EdgePred

**Figure 5: Comparison of PGMs and GraphPAR in the size of parameters tuned.**

## 6.4 Efficiency Analysis (RQ4)

We demonstrate the parameter efficiency of GraphPAR by comparing the parameters tuned to PGMs. As the results are shown in Table 5, the size of tuned parameters in GrahpPAR is 91% smaller than the size of parameters in the PGM. By contrast, since the parameter of the GNN encoder has to be tuned in traditional fair representation learning methods, the size of tuned parameters would be equal to or even larger than the size of PGM, far exceeding that in GraphPAR. In conclusion, GraphPAR is super parameter-efficient, which is well-suited for PGMs.

## 7 CONCLUSION

In this work, we explore the fairness in PGMs for the first time and discover that PGMs may capture more sensitive attribute semantics during the pre-training phase. In order to address this problem, we propose GraphPAR to endow PGMs with fairness during the adaptation for downstream tasks. Furthermore, based on the GraphPAR, we provide theoretical guarantees for fairness. Extensive experiments on real-world datasets demonstrate the effectiveness of GraphPAR in achieving fair predictions and providing provable fairness. In the future, we will further explore the research of PGMs in other trustworthy directions.

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

# A THE ALGORITHM OF GRAPHPAR

---

**Algorithm 1:** GraphPAR

---

**Data:** Graph $\mathcal{G} = (\mathcal{V}, \mathcal{E}, \mathbf{X})$, pre-trained graph model $f$
**Result:** Adapter $g$ and classifier $d$, and the provable fairness
  of each node

---

1 **1. GraphPAR Training:**
2 **for** *each epoch* **do**
3 ⎸ Compute the sensitive semantic vector as Eq 3;
4 ⎸ Sample the augmentation set $\hat{S}_i$ for each node $i$ as Eq 6;
5 ⎸ **if** *Train with RandAT* **then**
6 ⎸ ⎸ $\mathcal{L} = \mathcal{L}_{cls} + \lambda \mathcal{L}_{RandAT}$
7 ⎸ **else**
8 ⎸ ⎸ $\mathcal{L} = \mathcal{L}_{cls} + \lambda \mathcal{L}_{MinMax}$
9 ⎸ **end**
10 ⎸ Backward pass with $\mathcal{L}$;
11 **end**

12 **2. Provide Provable Fairness with Smoothing:**
13 Do adversarial training on the classifier $d$;
14 Construct the smoothed adapter $\widehat{g}$ and the smoothed
  classifier $\widehat{d}$;
15 **for** *each node $i$ in $\mathcal{V}$* **do**
16 ⎸ Compute the guarantee $d_{cs,i}$ of the adapter as
  ⎸ Theorem 1;
17 ⎸ Compute the guarantee $d_{rs,i}$ of the classifier as
  ⎸ Theorem 2;
18 ⎸ If $d_{cs,i} < d_{rs,i}$, then node $i$ has a provable fairness;
19 **end**

---

In this section, we describe the whole process of GraphPAR in Algorithm 1. GrapPAR consists of two parts: 1) train the adapter and classifier with RandAT or MinMax; 2) guarantee the fairness.

# B HYPERPARAMETER SENSITIVITY ANALYSIS

In this section, we conduct a more detailed hyperparameter sensitivity analysis for GraphPAR, focusing on three key aspects: the augmentation range, denoted as $\epsilon$, the augmentation sample number, represented by $k$, and the fairness loss scale, symbolized as $\lambda$. Each of these hyperparameters plays a crucial role in shaping the performance and fairness of GraphPAR, and understanding their sensitivity is vital for finding the best model for performance and fairness.

**Augmentation range sensitivity ($\epsilon$).** The augmentation range $\epsilon$ dictates the range of linear interpolation on sensitive attribute semantics. Within a certain range, the larger the augmentation range $\epsilon$, the larger the range of sensitive attributes considered, and the model fairer. For example, as depicted in Figure 7 (a), when the PGM is DGI and the debiasing method is MinMax, the metrics of DP and EO tend to decrease with increasing $\epsilon$ on the Credit dataset.

**Fairness loss scale factor sensitivity ($\lambda$).** $\lambda$ is a scale factor with respect to the fairness loss, which is used to balance accuracy and fairness. We find that different pre-training methods require

different values of $\lambda$. As depicted in Figure 6, for example, when the PGM is DGI, the optimal $\lambda$ is 0.7 in the Pokec_z and Credit datasets. However, the optimal $\lambda$ is 0.2 when the PGM is EdgePred.

**Augmentation sample number sensitivity ($k$).** $k$ is the augmentation sample number for each node representation. According to Figure 8 and Figure 9, we find that the optimal $k$ is associated with the dataset, the pre-training method, and the adapter training strategy, but the general RandAT requires a larger $k$ value than MinMax.

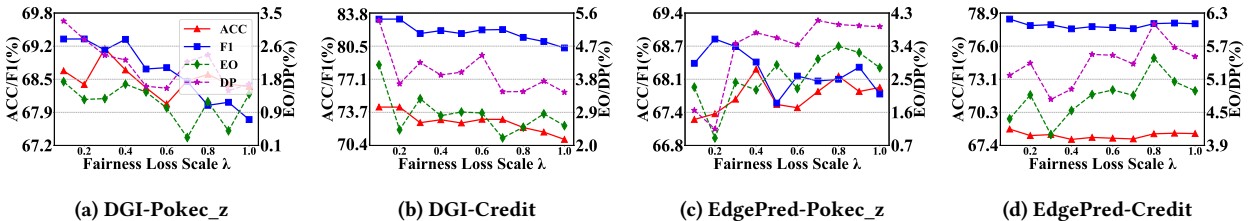

Figure 6: The effect of fairness loss scale factor $\lambda$ to GraphPAR$_{minmax}$.

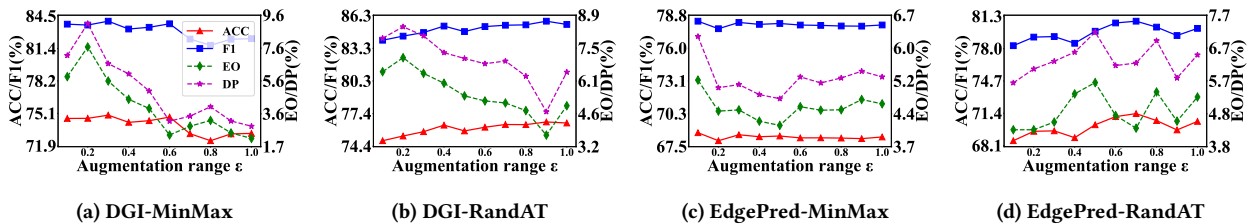

Figure 7: The effect of augmentation range $\epsilon$ to GraphPAR$_{minmax}$ and GraphPAR$_{RandAT}$ in the Credit dataset.

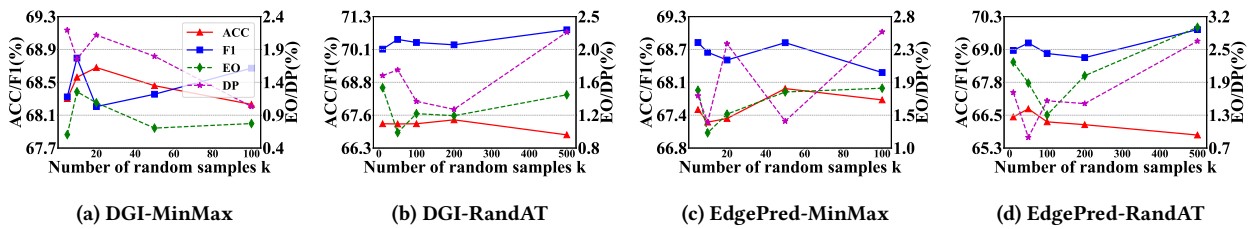

Figure 8: The effect of augmentation sample number $k$ to GraphPAR$_{minmax}$ and GraphPAR$_{RandAT}$ in the Pokec_z dataset.

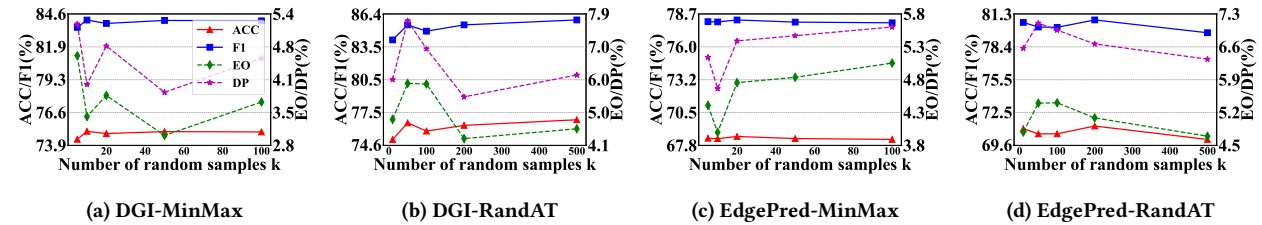

Figure 9: The effect of augmentation sample number $k$ to GraphPAR$_{minmax}$ and GraphPAR$_{RandAT}$ in the Credit dataset.

