# OpenReview forum: "Endowing Pre-trained Graph Models with Provable Fairness"
_ACM.org/TheWebConf/2024/Conference — TheWebConf24_

### Official Review · Reviewer_DFYL · 2023-11-24

**Novelty:** 5
**Technical Quality:** 6

**Review:**

This paper addresses the fairness issue in pre-trained graph models, highlighting issues in existing Pre-trained Graph Models (PGMs) that either demonstrate parameter inefficiency or lack theoretical guarantees. The authors introduce a novel approach, GraphPAR (Graph models with Provable fAiRness), which incorporates a parameter-efficient adapter with two adversarial debiasing methods to facilitate efficient parameter training. Furthermore, they offer theoretical assurances for fairness in training. Experiments are done on three datasets to demonstrate the effectiveness of GraphPAR.

Pros: The paper is well-written and logically structured. The idea of GraphPAR is somehow novel to me, addressing previously unexplored gaps related to fairness in graph models. The combination of theoretical justification and empirical evidence strengthens the credibility of the findings.

Cons: The paper could benefit from a more in-depth discussion of the methods used. Moreover, bringing in some visualized demonstrations, such as t-SNE, would aid in illustrating complex data and results, improving readability.

**Questions:**

The paper is well-organized and the motivation behind the study is clear. During my reading, I have several questions that I think the authors could address to further clarify the study's findings and implications.

1. In Section 4.1, an empirical analysis is presented to justify the derivation of $\alpha$. However, the explanation appears somewhat intuitive. Would visualizing the embeddings of $\alpha$, $h_{pos}$, and $h_{neg}$ provide a more concrete understanding of the additive performance in the embedding space?

2. Regarding the experimental results, it is common to observe a trade-off between fairness and accuracy. However, GraphPAR_{RandAT} not only improves fairness metrics but also enhances accuracy on the Credit dataset compared to DGI. Could you delve deeper into the reasons behind this outcome?

3. The introduced adapters are designed for parameter-efficient fine-tuning, yet normally their scalability seems to be more limited compared to full fine-tuning. How would the models perform if full fine-tuning were employed instead of using adapters? This information could be valuable in deciding between effectiveness and efficiency.

**Reviewer Confidence:**

3: The reviewer is confident but not certain that the evaluation is correct

**Scope:**

4: The work is relevant to the Web and to the track, and is of broad interest to the community

---

### Official Review · Reviewer_WAz4 · 2023-11-27

**Novelty:** 4
**Technical Quality:** 5

**Review:**

Summary

This paper works on fairness of pretrained graph models (PGMs). It first defines sensitive attribute semantic vector s, and then proposes adversarial learning methods which uses data augmented by scaled s. Theoretical results show that, with their assumptions, it can be highly likely the predictions made from h and h' will be the same. Experimental results show the proposed methods can improve EO and DP while maintaining high utility.

Strength

- This work proposes a simple method which works well in experiments.
- The paper is well presented and easy to read, except few confusing points.

Weakness

- I feel the experiment to verify whether \alpha satisfies expectation is not properly designed. Ideally, we should have ground truth of \alpha and compare the learned \alpha with the ground truth, which is probably better to be done with syntheic data where we are given the ground truth of \alpha. The classfication results shown in Fig. 3 only tell us moving the features towards the direction of \alpha will make the classifier not work. I doubt what will happen if we move the features towards any random directions, which should be added as a baseline.

- Theorem 1 basically says, if the distance ||h-h'||_2 is small enough, then one can guarantee their distance after adaptation by \hat{g}. This is a general known result with any smooth function \hat{g}, which is not really motivating Eq.(9) as h_i' is already randomly selected from a predefined set.

- Theorem 2 is also a property of smooth function, the connection between it and the methodology of this paper should be clarified.

**Questions:**

- What does S_h \leftarrow s mean exactly? Does it have to be adding s to the node / graph representation?

- Notation is confusing. \alpha denotes sensitive semantics in Sec 4, but in Sec 5.3, it becomes a scalar (probability).

- L549, how strong is the assumption that half of the points z are in the minimum enclosing ball g(h+N(0,\sigma_{cv}^2I))?

- g is not in Eq.(7), how does it connect to the RandAT method?

- How to choose the augmentation range \epsilon in practice? Can we select it by a validation set?


- The proof of Theorem 3 starts with an assumption that it holds for node i, how can one prove a Theorem by assuming the Theorem holds already?

**Reviewer Confidence:**

3: The reviewer is confident but not certain that the evaluation is correct

**Scope:**

4: The work is relevant to the Web and to the track, and is of broad interest to the community

---

### Official Review · Reviewer_bFPr · 2023-11-28

**Novelty:** 5
**Technical Quality:** 5

**Review:**

The paper looks at debiasing pre-trained graph models (PGMs). In particular, in a pre-trained graph model, the node embeddings can capture sensitive features of the original dataset, and thus leading to discriminatory behavior in downstream applications. While there are existing debiasing processes for GNNs, they don't readily apply to PGMs since these methods mostly involve parameter optimization, and thus cannot be efficiently carried out (i.e. reoptimize the PGM) for each new downstream task. The author(s) use a new approach based on augmenting the node embeddings from the PGM by different sensitive attribute semantics of the nodes, and then train the adapter and classifier of the downstream task to achieve fairness. The paper also specifies a way to provide provable fairness certification of the node embeddings. In the empirical evaluation, the author(s) compare to various fair adaptations of GNNs as well as their method (GraphPAR) for several common PGMs, and demonstrate considerably better fairness measure (both demographic parity and equality ipportunity) achieved by their method while performing on par with the best in terms of classification quality (by accuracy and F1). I think the overall methodology is sound and the analysis is well-written.

**Questions:**

No questions.

**Reviewer Confidence:**

1: The reviewer's evaluation is an educated guess

**Scope:**

3: The work is somewhat relevant to the Web and to the track, and is of narrow interest to a sub-community

---

### Official Review · Reviewer_g89Z · 2023-11-29

**Novelty:** 3
**Technical Quality:** 3

**Review:**

Quality: The ideas involved in the paper are sound, as is the goal of exploring the fairness of pre-trained graph models. The theoretical guarantees for “provable fairness” are a welcome addition. However, some parts do not entirely make sense, or their rationale is under-developed, as I’ve asked in the questions section. The experiments are fair, but need to be done over more datasets. Two datasets are already of a similar kind here.

Clarity: The paper requires more polishing before it can be accepted for publication. The language is hard to read in multiple places, and sometimes difficult to understand what the authors want to convey. Some of the mathematical equations are also ill-formed, e.g., equation 2 where they probably meant $\forall \boldsymbol{s}, \boldsymbol{s}’ \text{s.t.} ||\boldsymbol{s} - \boldsymbol{s}’||_2 \ne 0$.

Significance: The idea is somewhat significant as it is applicable to all kinds of pre-trained graph models in general. A plug-and-play adapter component would be useful if it could mitigate fairness and bias issues without having to touch the core of training, i.e., without having to be applied in the loss function.

Originality: The authors use already well-established techniques in their work. The ideas of adapter tuning and adversarial debiasing are well-known. Their application to this domain is somewhat novel, and it is not difficult to conceive an adapter on top of a PGM as a black-box, although it seems useful.

**Questions:**

- Could you experiment on more graph datasets of a diverse nature, so as to concretize your results?

- Could you explain better the rationale behind how you compute the semantics vector $\boldsymbol{\alpha}$? What does it mean to compute the “average” of latent embeddings of positive or negative examples - what do they represent intuitively? Why perform a subtraction between these two averages - what does it signify?

**Reviewer Confidence:**

3: The reviewer is confident but not certain that the evaluation is correct

**Scope:**

2: The connection to the Web is incidental, e.g., use of Web data or API

---

### Official Review · Reviewer_hhDi · 2023-11-30

**Novelty:** 5
**Technical Quality:** 6

**Review:**

The subject of this paper is pre-trained graph models (PGMs) and to what extend one can ensure their "fairness" in downstream tasks. Indeed, as also shown by the authors, PGMs are able to capture sensitive attribute semantics during the pre-training phase and, as a result, be even more unfair in downstream applications than standard Graphical Convolutional Networks.

The paper proposes a framework for endowing PGMs with provable fairness. In particular, the proposed method first augments the representation of sensitive attributes for each node. This is done via a parameter-efficient adapter that ensures fairness of the resulting representations. The adapter is trained via two adversarial debiasing methods (random augmentation adversarial training and min-max adversarial attacks training) which guarantee both fairness and performance of downstream tasks.

Finally, the authors provide theoretical guarantees for fairness and extensive experiments on real-world datasets to demonstrate the effectiveness of their approach in comparison to other baselines.

Overall, I feel that the contributions of the paper (both on a technical and a conceptual level) are above the bar of acceptance of this conference.

**Questions:**

Could you comment on why do you think the performance of GraphPAR varies among different PGMs or, in other words, what is about DGI that makes it more suitable for applying your method?

**Reviewer Confidence:**

2: The reviewer is willing to defend the evaluation, but it is likely that the reviewer did not understand parts of the paper

**Scope:**

4: The work is relevant to the Web and to the track, and is of broad interest to the community

---

### Decision · Program_Chairs · 2024-01-22

**Decision:**

Accept

**Comment:**

Note: Authors raised concerns about reviewer g89Z (who gave the lowest ratings) being uncommunicative. My recommendation of weak accept (leaning borderline) is consistent with my own reading of the paper and reviews, with or without the authors explicitly raising the concern.

 This paper proposes a method to add an adapter onto a fixed, already-trained graph model in order to improve fairness. The technique involves augmenting the representation of sensitive attributes for each node, which is interesting in its own right.

 The reviews are generally positive, with some issues around clarity. It studies an interesting problem and proposes a mathematically justified solution. It's the first paper (to my knowledge) that attempts to address fairness issues on a fixed, pretrained graph model; for this reason, it's interesting and novel. However, some more motivation for why this is important would be good. (The authors do note that downstream applications may have different notions of what attributes are sensitive, which is a reasonable start.) The mathematical understanding is useful, although there are more mathematically-rich analyses of fairness.

 Strengths:
 * Investigates a new method to handle fairness, built to work with pretrained models.
 * Provides a simple mathematical analysis as well as experiments.

 Weaknesses:
 * Needs more motivation around applicability. In particular, the definition of fairness suggests that we should simply ignore the value of sensitive features. (I'm not certain I agree, but this is a simple and established notion.) So one valid technique is to simply retrain the GNN without sensitive features. It might be the case that this retraining is expensive, and the authors note that a single PGM may be used with different sensitive attributes (which would make this impractical). But it would be good to have more motivation for why fixing PGMs is so valuable.
 * Some issues with clarity.